# STAT3 Inhibits CD103^+^ cDC1 Vaccine Efficacy in Murine Breast Cancer

**DOI:** 10.3390/cancers12010128

**Published:** 2020-01-04

**Authors:** Taylor T. Chrisikos, Yifan Zhou, Haiyan S. Li, Rachel L. Babcock, Xianxiu Wan, Bhakti Patel, Kathryn Newton, James J. Mancuso, Stephanie S. Watowich

**Affiliations:** 1Department of Immunology, University of Texas MD Anderson Cancer Center, Houston, TX 77030, USA; TTChrisikos@mdanderson.org (T.T.C.); YZhou14@mdanderson.org (Y.Z.); haiyan.s.li@gmail.com (H.S.L.); RDziuk@mdanderson.org (R.L.B.); XWan2@mdanderson.org (X.W.); BPatel6@mdanderson.org (B.P.); kathrynnewton4@yahoo.com (K.N.); JJMancuso@mdanderson.org (J.J.M.); 2MD Anderson UTHealth Graduate School of Biomedical Sciences, Houston, TX 77030, USA

**Keywords:** CD103^+^ dendritic cells, cDC1, STAT3, breast cancer, immunotherapy, tumor vaccine, immunosuppression

## Abstract

Conventional dendritic cells (cDCs) are a critical immune population, composed of multiple subsets, and responsible for controlling adaptive immunity and tolerance. Although migratory type 1 cDCs (CD103^+^ cDC1s in mice) are necessary to mount CD8^+^ T cell-mediated anti-tumor immunity, whether and how tumors modulate CD103^+^ cDC1 function remain understudied. Signal Transducer and Activator of Transcription 3 (STAT3) mediates the intracellular signaling of tumor-associated immunosuppressive cytokines, such as interleukin (IL)-10; thus, we hypothesized that STAT3 restrained anti-tumor immune responses elicited by CD103^+^ cDC1s. Herein, we show that in vitro-derived STAT3-deficient (*Stat3*^∆/∆^) CD103^+^ cDC1s are refractory to the inhibitory effects of IL-10 on Toll-like receptor 3 (TLR3) agonist-induced maturation responses. In a tumor vaccination approach, we found *Stat3*^∆/∆^ CD103^+^ cDC1s restrained mammary gland tumor growth and increased mouse survival more effectively than STAT3-sufficient CD103^+^ cDC1s. In addition, vaccination with *Stat3*^∆/∆^ CD103^+^ cDC1s elicited increased amounts of tumor antigen-specific CD8^+^ T cells and IFN-γ^+^ CD4^+^ T cells in tumors and tumor-draining lymph nodes versus phosphate-buffered saline (PBS)-treated animals. Furthermore, IL-10 receptor-deficient CD103^+^ cDC1s controlled tumor growth to a similar degree as *Stat3*^∆/∆^ CD103^+^ cDC1s. Taken together, our data reveal an inhibitory role for STAT3 in CD103^+^ cDC1 maturation and regulation of anti-tumor immunity. Our results also suggest IL-10 is a key factor eliciting immunosuppressive STAT3 signaling in CD103^+^ cDC1s in breast cancer. Thus, inhibition of STAT3 in cDC1s may provide an important strategy to improve their efficacy in tumor vaccination approaches and cDC1-mediated control of anti-tumor immunity.

## 1. Introduction

Conventional dendritic cells (cDCs) are the primary antigen-presenting cells (APCs) of the immune system [1]. The cDCs comprise type 1 (cDC1) and type 2 (cDC2) subsets. Each population is further subdivided by anatomical location, including distinct tissue-resident migratory and lymphoid organ-residing subsets [2]. Migratory cDCs are responsible for surveying peripheral tissues at steady state, promoting tolerance [3,4]. In addition, cDCs express an array of receptors that bind to danger- and pathogen-associated molecular patterns including Toll-like receptors (TLRs). Upon TLR activation, DCs upregulate major histocompatibility molecule (MHC) I and MHC II, T cell co-stimulatory molecules CD80 and CD86, and inflammatory cytokines and chemokines in a process termed DC maturation [1]. Migratory DCs that undergo maturation travel to the lymph nodes (LNs); within LNs, migratory DCs present tissue-derived antigens to naïve lymphocytes and initiate adaptive immune responses [1].

Extracellular antigens such as tumor or bacterial products are presented by DCs via distinct processes. The direct presentation pathway comprises MHC II antigen display to CD4^+^ T cells, thus stimulating CD4^+^ T cell responses. By contrast, DC presentation of extracellular antigen on MHC I occurs by the antigen cross-presentation pathway. This results in DC-mediated activation of CD8^+^ T cells, the primary cytotoxic subset of the adaptive immune response. The cDC1s are the most efficient DC subset at antigen cross-presentation and CD8^+^ T cell activation [5]. In agreement, murine migratory cDC1s (CD103^+^ cDC1s) are necessary for the generation of CD8^+^ T cell-mediated anti-tumor immunity [6,7,8]. Moreover, CD103^+^ cDC1s are a major source of CD8^+^ T cell chemoattractants, including C-X-C motif chemokine ligand 9 (CXCL9) and CXCL10, in the tumor microenvironment (TME); these chemokines drive T cell recruitment and anti-tumor immunity [9,10]. Consistent with these findings, intratumoral accumulation of CD103^+^ cDC1s, or the human equivalent CD141^+^ cDC1s, is positively correlated with overall patient survival and predicts responsiveness to immunotherapy [10,11,12,13].

Immune checkpoint blockade (ICB) and CD8^+^ T cell-based immunotherapies have demonstrated great success in the treatment of cancer [14,15]. ICB in particular is associated with long-term, durable responses, suggesting establishment of anti-tumor immune memory. Despite these powerful advances, many types of cancer, such as breast cancer, remain non-responsive to immunotherapy [16]. Resistance to ICB is thought to be, in part, due to a lack of pre-existing anti-tumor T cells in the TME resulting from an immunosuppressive microenvironment [14]. Therefore, further study is required to increase our understanding of immune–TME interactions and generate novel therapeutic strategies that improve the ability of the immune system to overcome TME immunosuppression and effectively treat cancer.

The ability of DCs to induce the adaptive immune response has led to investigations of their use as an anti-tumor treatment; however, DC-based therapies have demonstrated modest efficacy in clinical trials thus far [17,18,19]. While the underlying causes of poor patient responses are unclear, new information points to potential contributing factors that may affect DC therapeutic efficacy. The delineation of unique functional roles for specific DC subsets suggests that utilization of suboptimal DC populations may contribute to modest treatment response. For instance, several clinical trials have utilized DCs of monocytic origin (monocyte-derived DCs, moDCs) [17]. In recent work, we investigated the anti-tumor efficacy of in vitro-generated CD103^+^ cDC1s, which resemble the migratory cDC1 subset found in vivo based on expression of canonical transcriptional regulators and cell surface markers, and efficient antigen cross-presentation to CD8^+^ T cells upon stimulation with polyinosinic-polycytidylic acid (poly I:C) [20]. Not only did in vitro-generated CD103^+^ cDC1s induce potent anti-tumor immunity, they demonstrated increased efficacy when compared with moDCs [20]. Moreover, multiple factors within the TME, including immunosuppressive cytokines, inhibit DC function [9,21,22,23]. Collectively, these data suggest strategies utilizing cDC1s that are refractory to tumor-associated inhibitory factors may overcome treatment barriers and provide new approaches for cancer immunotherapy.

Signal Transducer and Activator of Transcription 3 (STAT3) mediates intracellular signaling elicited by immunosuppressive and anti-inflammatory cytokines, such as interleukin (IL)-10. Depletion of *Stat3* in CD11c^+^ cells, including all DC subsets, alters immune homeostasis, leading to colon inflammation and increased serum amounts of inflammatory cytokines such as IL-12 and interferon gamma (IFN-γ) [24]. In addition, STAT3 is well known for its immunomodulatory activity in cancer [25]. For instance, *Stat3* deletion from hematopoietic cells results in increased anti-tumor immunity [22]. Moreover, myeloid-restricted *Stat3* deletion leads to enhanced production of IL-12 and reduced expression of tumor-promoting cytokines such as IL-23 and IL-10 [23]. Previously, we showed that *Stat3* deletion in moDCs used as a tumor vaccine enhances accumulation of effector CD8^+^ and CD4^+^ T cells, and suppresses FoxP3^+^ CD4^+^ T cell (Treg) accrual in melanoma tumors, correlating with decreased tumor growth and increased mouse survival [21]. Whether STAT3 regulates CD103^+^ cDC1 function in the TME, however, is unclear. This is important to address as CD103^+^ cDC1s have promise as a next generation DC-based tumor immunotherapy [20]; understanding factors regulating CD103^+^ cDC1 function could lead to novel immunotherapeutic approaches.

Here, we examined whether STAT3 controlled the efficacy of CD103^+^ cDC1s used as a cell-based vaccine for murine breast cancer, a cancer type that is refractory to immunotherapy [16]. Importantly, use of this vaccine approach allowed us to circumvent the lack of selective genetic models to delete *Stat3* from CD103^+^ cDC1s in vivo. Our results indicate an inhibitory role for STAT3 in CD103^+^ cDC1-mediated anti-tumor immunity, including STAT3-dependent suppression of cDC1 maturation and IFN-γ^+^ T cell responses. These results suggest efforts to modulate STAT3 function in tumor-associated cDC1s may improve their ability to elicit anti-tumor immunity.

## 2. Results

### 2.1. STAT3 Is Required for IL-10-Mediated Inhibition of CD103^+^ cDC1 Maturation

To examine STAT3 function in the murine CD103^+^ cDC1 subset, we utilized a bone marrow culture system that provides efficient production of CD103^+^ cDC1s in vitro [26]. Our prior studies show in vitro-generated CD103^+^ cDC1s possess key features of the nonlymphoid organ cDC1 subset found in vivo, and serve as an effective tumor vaccine in murine melanoma and osteosarcoma [20]. We generated CD103^+^ cDC1s from bone marrow cells isolated from CD11c Cre^+^
*Stat3^fl/fl^* (*Stat3*^∆/∆^) or CD11c Cre^−^
*Stat3^fl/fl^* (*Stat3^fl/fl^*) mice (Appendix A). The CD11c Cre^+^ model was selected as we anticipated that Cre-driven *Stat3* deletion from CD11c^+^ cells would not affect generation of CD103^+^ cDC1s, since STAT3 is dispensable for homeostatic production of this population in vivo [21,27,28,29]. Consistent with this expectation, we confirmed STAT3 depletion from *Stat3*^∆/∆^ CD103^+^ cDC1 cultures, although residual amounts of tyrosine-phosphorylated STAT3 (pSTAT3) were detected upon IL-10 stimulation, indicating efficient but not complete STAT3 removal (Appendix A). Furthermore, we found similar expansion of CD103^+^ cDC1 numbers as well as total cell amounts in *Stat3*^∆/∆^ and *Stat3^fl/fl^* bone marrow cultures in vitro (Appendix A). Expression of cDC1-related transcriptional regulators *Batf3, Id2,* and *Irf8,* as well as plasmacytoid DC (pDC)-related factors *Tcf4* and *Zeb2,* were also comparable between *Stat3*-sufficient and -deficient CD103^+^ cDC1s (Appendix A). Taken together, these results indicate that targeted *Stat3* deletion in CD11c^+^ cells allows for efficient production of *Stat3*^∆/∆^ CD103^+^ cDC1s in vitro, suggesting STAT3 is dispensable for terminal stages of CD103^+^ cDC1 development. 

We next investigated whether STAT3 controlled the expression of key DC maturation markers involved in T cell activation. To test this, we evaluated steady state expression, as well as expression upon stimulation with the TLR3 agonist poly I:C in the absence or presence of the immunomodulatory cytokine IL-10, which is known to elicit STAT3 signaling [30]. MHC I and II, and CD40 were expressed similarly in *Stat3^fl/fl^* and *Stat3*^∆/∆^ CD103^+^ cDC1s under all conditions (Appendix A). Poly I:C treatment enhanced cell surface expression of CD80 and CD86 as expected, and these co-stimulatory molecules were induced to comparable amounts in *Stat3^fl/fl^* and *Stat3*^∆/∆^ CD103^+^ cDC1s (Figure 1A). By contrast, simultaneous incubation with IL-10 suppressed poly I:C-mediated induction of CD80 and CD86 in *Stat3^fl/fl^* CD103^+^ cDC1s, while *Stat3*^∆/∆^ CD103^+^ cDC1s maintained high amounts of CD80 and CD86 in these conditions (Figure 1A). In addition, poly I:C-mediated induction of inflammatory cytokine and chemokine expression was inhibited by IL-10 in *Stat3^fl/fl^* but not *Stat3*^∆/∆^ CD103^+^ cDC1s (Figure 1B, Appendix A). These results indicate IL-10 signals via STAT3 to suppress key maturation responses of CD103^+^ cDC1s upon poly I:C stimulation.

### 2.2. CD103^+^ cDC1 Vaccine Efficacy in Breast Cancer Is Suppressed by STAT3

To examine whether STAT3 regulates the anti-tumor activity of CD103^+^ cDC1s, we used an established intratumoral (i.t.) vaccination strategy [20]. We selected a breast cancer model for these studies, as this prevalent cancer is refractory to other immunotherapeutic approaches [16]. Mice bearing bilateral orthotopic Polyomavirus middle T-antigen (PyMT) mammary tumors expressing the surrogate tumor antigen ovalbumin (PyMT-OVA) were vaccinated i.t. with *Stat3^fl/fl^* or *Stat3*^∆/∆^ CD103^+^ cDC1s, following DC exposure to maturation stimuli and OVA. Animals were vaccinated i.t. on the right side 7 days (d) after PyMT-OVA tumor establishment, while the left side tumors remained untreated. This approach allows for assessment of local and systemic anti-tumor responses in the vaccinated and unvaccinated tumors, respectively [20].

The *Stat3*-sufficient CD103^+^ cDC1 vaccine restrained bilateral mammary tumor growth (Figure 2A,B), consistent with our prior observations in murine melanoma and osteosarcoma models [20]. Vaccination with *Stat3*^∆/∆^ CD103^+^ cDC1s resulted in improved control of bilateral tumors relative to *Stat3^fl/fl^* CD103^+^ cDC1 vaccination or phosphate buffered saline (PBS)-treated controls (Figure 2A,B). Mouse median survival was also enhanced upon vaccination with *Stat3^fl/fl^* CD103^+^ cDC1s, and further improved by the *Stat3*^∆/∆^ CD103^+^ cDC1 vaccine (Figure 2C). In addition, vaccination with *Stat3*^∆/∆^ CD103^+^ cDC1s led to decreased mean tumor mass in vaccinated but not non-vaccinated tumors 10 days after delivery of the therapy, compared to PBS-treated controls (Figure 2D). By contrast, mean tumor mass was indistinguishable between treatment groups 4 days following vaccination (Figure 2D), suggesting initial tumor growth was comparable. Collectively, these results indicate that STAT3 inhibits the capacity of an in vitro-generated CD103^+^ cDC1 vaccine to induce systemic anti-tumor immunity in murine breast cancer.

### 2.3. STAT3 Restrains Co-Stimulatory Molecule Expression in CD103^+^ cDC1s Post-Vaccination

To evaluate mechanisms by which *Stat3*^∆/∆^ CD103^+^ cDC1s improved control of PyMT mammary tumors, we examined the maturation state of CD103^+^ cDC1s post-vaccination. Congenic CD45.1^+^ mice bearing bilateral PyMT-OVA tumors were vaccinated with CD45.2^+^
*Stat3^fl/fl^* or *Stat3*^∆/∆^ CD103^+^ cDC1s, and subsequently analyzed for vaccine-derived CD45.2^+^ CD103^+^ cDC1s 40 hours (h) following vaccination. We found increased expression of CD86 on tumor-associated, vaccine-derived *Stat3*^∆/∆^ CD103^+^ cDC1s compared to *Stat3^fl/fl^* CD103^+^ cDC1s, as judged by relative CD86 amounts on CD45.2^+^ CD103^+^ cDC1s (Figure 3A,B). In addition, CD80 and MHC II appeared to be increased on vaccine-derived *Stat3*^∆/∆^ CD103^+^ cDC1s in the TME, relative to vaccine-derived *Stat3^fl/fl^* CD103^+^ cDC1s, although these data did not reach statistical significance (Figure 3A,B and Appendix A). By contrast, STAT3 did not affect expression of CD86, CD80, or MHC II on vaccine-derived CD103^+^ cDC1s within tumor-draining lymph nodes (TdLNs) (Figure 3A,B and Appendix A). Furthermore, STAT3 did not alter the amounts of vaccine-derived CD103^+^ cDC1s in mammary tumors or TdLNs, as judged by equivalent numbers of CD45.2^+^
*Stat3^fl/fl^* and CD45.2^+^
*Stat3*^∆/∆^ CD103^+^ cDC1s detected in each site (Figure 3C,D). Notably, *Stat3^fl/fl^* and *Stat3*^∆/∆^ CD45.2^+^ CD103^+^ cDC1s were found only in the vaccinated tumors and corresponding TdLNs (Appendix A), suggesting they did not migrate to the unvaccinated tumor or distal TdLN. Taken together, these results indicate STAT3 inhibits CD86 expression in CD103^+^ cDC1s within the murine breast TME but does not affect CD103^+^ cDC1 retention in tumors or CD103^+^ cDC1 migration to TdLNs.

### 2.4. CD103^+^ cDC1 Vaccine-Mediated Increase of Tumor Antigen-Specific CD8^+^ T Cells and IFN-γ^+^ CD4^+^ T Cells Is Inhibited by STAT3

CD103^+^ cDC1 vaccination i.t. in murine melanoma or osteosarcoma promotes selective accumulation of IFN-γ^+^ CD4^+^ T cells (T helper 1, Th1) and IFN-γ^+^ CD8^+^ T cells within tumors [20]. To test whether CD103^+^ cDC1 vaccination induces similar responses in murine breast cancer, and if STAT3 modulates CD103^+^ cDC1-mediated immune responses, we evaluated immune cell populations within tumors and TdLNs following vaccination of mice bearing bilateral PyMT-OVA tumors with *Stat3^fl/fl^* or *Stat3*^∆/∆^ CD103^+^ cDC1s. We assessed immune infiltrates at 4 d and 10 d post vaccination. Analyzing at d 4 allows for the assessment of early immune events after vaccination, before differences in tumor size become apparent, while at d 10 vaccinated tumors begin to regress (Figure 2D), which may influence immune infiltration. Upon vaccination with *Stat3*-sufficient CD103^+^ cDC1s we observed a trend toward increased amounts of tumor antigen (OVA)-specific CD8^+^ T cells, as indicated by analysis of SIINFEKL/H-2Kb pentamer^+^ CD8^+^ T cells, and total Th1 cells within tumors or TdLNs at d 4 post-vaccination, relative to PBS-treated controls, although these increases were not significant statistically (Figure 4). By d 10 post-vaccination, OVA-specific CD8^+^ T cells were reduced in TdLNs corresponding to *Stat3^fl/fl^* CD103^+^ cDC1-vaccinated tumors compared to the PBS treatment group, but were unaffected in the distal TdLN or the tumors and TdLNs of the *Stat3*^∆/∆^ CD103^+^ cDC1 vaccine-treated cohort (Figure 4A,B). By contrast, *Stat3*^∆/∆^ CD103^+^ cDC1s stimulated accumulation of OVA-specific CD8^+^ T cells in vaccine-treated tumors and corresponding TdLNs at 4 d post-vaccination (Figure 4A,B). In addition, *Stat3*^∆/∆^ CD103^+^ cDC1-vaccinated tumors and respective TdLNs had increased numbers of Th1 cells 4 d following vaccination, compared to PBS-treated mice (Figure 4C,D). Furthermore, elevated amounts of Th1 cells were found in non-vaccinated tumors of mice that received the *Stat3*^∆/∆^ CD103^+^ cDC1 vaccine at 4 d post-vaccination, and showed a persistent increase in tumors of this cohort, as indicated by their detection 10 d following vaccination (Figure 4C,D). *Stat3*^∆/∆^ CD103^+^ cDC1-vaccinated tumors also appeared to have increased amounts of total IFN-γ^+^ CD8^+^ T cells, as well as CD8^+^ T cell to Treg ratios, compared to *Stat3^fl/fl^* CD103^+^ cDC1-vaccinated and PBS-treated mice, although these data were not significant statistically (Appendix A). Myeloid populations were not affected by vaccination with *Stat3^fl/fl^* or *Stat3*^∆/∆^ CD103^+^ cDC1s (Appendix A). Taken together, these results indicate STAT3 inhibits the ability of the CD103^+^ cDC1 vaccine to induce tumor-antigen specific CD8^+^ T cell- and Th1 cell-mediated immunity in the murine breast TME and TdLNs.

### 2.5. Signaling via the IL-10 Receptor Inhibits CD103^+^ cDC1 Vaccine Efficacy

Our results indicate STAT3 signaling within CD103^+^ cDC1s elicits immunosuppressive responses including restrained expression of T cell co-stimulatory molecules and cytokines, as well as suppressed CD103^+^ cDC1-mediated increases in tumor antigen-specific CD8^+^ T cell and Th1 cell subsets in vivo (Figure 1, Figure 3A,B, Figure 4 and Appendix A). The extracellular factors capable of activating STAT3 in CD103^+^ cDC1s within the murine breast TME, however, remained unclear. To address this, we performed cytokine multiplex analysis of PyMT-OVA culture supernatants. These experiments revealed numerous cytokines and chemokines produced by PyMT-OVA cells, including the STAT3-activating factors IL-10, IL-6, leukemia inhibitory factor (LIF), and granulocyte colony-stimulating factor (G-CSF) (Figure 5A). CD103^+^ cDC1s express relatively low amounts of *Lifr* and *Csf3r* mRNAs versus other immune subsets such as pDCs or granulocytes, respectively [31], suggesting they may be refractory tumor-produced LIF or G-CSF. By contrast, mRNA expression of the unique receptor subunits for IL-6 (*Il6ra*) and IL-10 (*Il10rb*) are more abundant in CD103^+^ cDC1s [31]. Previous work by others has shown IL-10 suppresses induction of IFN-γ from CD4^+^ T cells during co-culture with a macrophage cell line [32]. IL-10 also restrains moDC-mediated CD8^+^ T cell proliferation and cytotoxic activity [33]. As these responses parallel roles we identified for STAT3 in CD103^+^ cDC1s, we evaluated whether IL-10 signaling within CD103^+^ cDC1s inhibits vaccine efficacy in murine PyMT-OVA tumors.

To block IL-10 signaling in CD103^+^ cDC1s specifically, we utilized bone marrow from *Il10rb*^−/−^ mice for generation of CD103^+^ cDC1s in vitro. The *Il10rb*^−/−^ animals lack expression of the IL-10 receptor β subunit (IL-10Rβ), which is essential for IL-10 receptor (IL-10R) signal transduction [34]. IL-10Rβ deficiency did not alter development of CD103^+^ cDC1s in culture (Appendix A). Moreover, *Il10rb*^−/−^ mice did not display alterations in total cDC, cDC1, or cDC2 amounts in lungs, liver, or spleen (Appendix A), suggesting IL-10Rβ is dispensable for cDC development.

We next evaluated the efficacy of in vitro-generated *Il10rb*^−/−^ CD103^+^ cDC1s in tumor vaccination assays in mice bearing bilateral PyMT-OVA tumors. These experiments showed that the *Il10rb*^−/−^ CD103^+^ cDC1 vaccine suppressed bilateral tumor growth effectively, and to a similar degree as that observed upon vaccination with *Stat3*^∆/∆^ CD103^+^ cDC1s (Figure 5B,C). Taken together, our data imply that IL-10-mediated signaling via STAT3 elicits immunosuppressive responses in CD103^+^ cDC1s, thereby reducing CD103^+^ cDC1-mediated anti-tumor immunity.

## 3. Discussion

CD103^+^ cDC1s are critical for the generation of CD8^+^ T cell-mediated anti-tumor immunity, and their intratumoral abundance positively correlates with patient survival and predicts response to immunotherapy [6,12,13]. The TME also contains immunosuppressive factors, including STAT3-activating cytokines [9,21,22,23], although their effects on CD103^+^ cDC1 function are understudied. Our data demonstrate STAT3 and IL-10R inhibit the efficacy of a CD103^+^ cDC1 vaccine in murine breast cancer. *Stat3*^∆/∆^ CD103^+^ cDC1- or *Il10rb*^−/−^ CD103^+^ cDC1-vaccinated mice exhibited delayed bilateral tumor growth and increased survival, compared to mice vaccinated with *Stat3^fl/fl^* CD103^+^ cDC1s. As non-vaccinated tumors also responded to the *Stat3*^∆/∆^ or *Il10rb*^−/−^ CD103^+^ cDC1 vaccine, this indicates promotion of systemic anti-tumor immunity. Consistently, improved vaccine efficacy with *Stat3*^∆/∆^ CD103^+^ cDC1s correlated with increased numbers of tumor-antigen specific CD8^+^ T cells and Th1 cells in tumors and TdLNs. Anti-tumor CD8^+^ T cell and Th1 cell abundance in the TME is crucial for immune-mediated clearance of tumors [35]. Thus, taken together, our results suggest STAT3 and IL-10R signaling in CD103^+^ cDC1s inhibits their ability to induce systemic anti-tumor adaptive immunity.

Previously, others have shown that breast tumors suppress cDC1 development from DC progenitors, resulting in decreased cDC1 abundance and reduced CD8^+^ T cell-mediated anti-tumor immunity [36]. The inhibition of cDC1 development by breast cancers is due to their production of G-CSF, a STAT3-activating cytokine [36]; consistent with these prior studies, we found PyMT tumor cells secrete G-CSF in culture. Our CD103^+^ cDC1 vaccination approach, therefore, allowed us to circumvent the inhibitory effects of PyMT-derived G-CSF on cDC1 development. Moreover, differentiated CD103^+^ cDC1s express only low amounts of *Csf3r* mRNA [31], suggesting they are refractory to tumor-produced G-CSF within the TME. Conversely, STAT3 is required for proliferation of DC progenitors in response to FMS-related tyrosine kinase 3 ligand (FLT3L), an important DC growth factor [27,37]. We did not detect differences in generation of *Stat3^fl/fl^* and *Stat3*^∆/∆^ CD103^+^ cDC1s in culture. These results are consistent with our prior studies, which showed STAT3 is dispensable for generation of CD103^+^ cDC1s in homeostasis [27], and collectively indicate STAT3 is not required for terminal differentiation of CD103^+^ cDC1s. Taken together, the data underscore specific roles for STAT3 in cytokine-mediated DC development, as well as the usefulness of our vaccination approach to directly examine STAT3 function in differentiated CD103^+^ cDC1s.

STAT3 is known to suppress the inflammatory responses of myeloid subsets [38]. For example, STAT3 inhibits TLR4-mediated signaling in bone marrow-derived macrophages (BMDMs) [39], as well as macrophage and neutrophil activity in vivo [40]. Consistently, mice with *Stat3* deletion in myeloid or DC populations develop intestinal inflammation, attributed to hyperreactivity of these subsets to gut-derived microbial products [24,41]. STAT3 deletion renders BMDMs or moDCs hyperactive to TLR stimulation, as determined by increased cytokine secretion upon TLR agonist treatment [24,39]. By contrast, we found CD103^+^ cDC1s required concomitant treatment with IL-10 to reveal the inhibitory activity of STAT3 on production of inflammatory factors and induction of co-stimulatory molecules following poly I:C stimulation. These results suggest distinct mechanisms elicit STAT3 anti-inflammatory function in macrophage or moDCs populations versus CD103^+^ cDC1s. In BMDMs, STAT3 anti-inflammatory activity was linked to STAT3-mediated suppression of the TLR signaling molecule Ubc13, and to BMDM production of IL-6, which elicited an autocrine anti-inflammatory response [39]. By contrast, STAT3 signaling in moDCs inhibited expression of the transcriptional regulator Id2, a pathway that suppressed moDC-mediated anti-tumor activity [21]. Further work is required to evaluate whether CD103^+^ cDC1s utilize similar or distinct STAT3 anti-inflammatory or anti-tumor mechanisms in response to TLR3 agonist stimulation.

Although we found poly I:C-activated CD103^+^ cDC1s produce a number of cytokines, these factors appear insufficient to elicit STAT3-mediated inhibitory activity in an autocrine fashion, since in vitro maturation responses were only repressed in *Stat3*-sufficient CD103^+^ cDC1s exposed to an exogenous source of IL-10. In addition, *Stat3*-sufficient and -deficient CD103^+^ cDC1s isolated from PyMT-OVA tumors but not TdLNs showed differences in co-stimulatory molecule expression. Taken together, these data suggest an external cytokine source(s) present in the TME is required to inhibit CD103^+^ cDC1 function via STAT3. We found PyMT-OVA cells produced the STAT3-activating cytokines IL-10, IL-6, LIF, and G-CSF. Moreover, we showed *Il10rβ*^−/−^ CD103^+^ cDC1s have increased vaccine efficacy, similar to that of *Stat3*^∆/∆^ CD103^+^ cDC1s. Taken together with results from prior work [24], our findings suggest an IL-10–STAT3 axis inhibits CD103^+^ cDC1 function in the TME to prevent anti-tumor adaptive immunity. While our results do not rule out inhibitory roles for tumor-expressed IL-6 on CD103^+^ cDC1 function, we anticipate CD103^+^ cDC1s are refractory to tumor-derived LIF or G-CSF due to the relatively low abundance of receptor mRNAs in this subset relative to other immune populations [31].

Recent reports indicate IL-10 is capable of stimulating the proliferation and activity of CD8^+^ T cells in the TME, responses that promote anti-tumor immunity [42,43]. In addition, a pegylated form of IL-10, with increased half-life compared to recombinant IL-10, activates anti-tumor CD8^+^ T cells in humans [44]. Thus, the current data suggests multiple and discrete roles for IL-10 in the TME, including suppressive effects on DC function and stimulatory activity toward CD8^+^ T cells. These results suggest approaches that target IL-10 to CD8^+^ T cells specifically may alleviate potential inhibitory effects of this cytokine on DCs, and thereby improve IL-10 treatment efficacy in cancer.

Our data revealed STAT3 restrains co-stimulatory molecule expression in CD103^+^ cDC1s in the TME, particularly CD86 cell surface amounts. In addition to roles in naïve T cell priming, CD80 and CD86 induce proliferation and IFN-γ expression in exhausted, tumor-specific CD8^+^ T cells within the TME following vaccination with moDCs [45]. Furthermore, CD28, the T cell receptor for CD80 and CD86, induces T cell proliferation post PD-1 blockade in chronic viral infection and in tumors [46]. Thus, increased CD86 expression in *Stat3*^∆/∆^ CD103^+^ cDC1s in the TME post vaccination may contribute to the increase in tumor antigen-specific CD8^+^ T cells. In addition, IFN-γ and CXCL10, which were induced by poly I:C but inhibited by IL-10 in a STAT3-dependent manner in CD103^+^ cDC1s, promote the function and recruitment of CD8^+^ T cells in the TME [10,47]. This suggests enhanced cytokine and chemokine expression may contribute to the increases of anti-tumor CD8^+^ T cells and Th1 cells in the TME and TdLNs of *Stat3*^∆/∆^ CD103^+^ cDC1 vaccinated mice. Interestingly, although tumor antigen-specific CD8^+^ T cells and Th1 cells were elevated in the TdLNs of *Stat3*^∆/∆^ CD103^+^ cDC1-vaccinated mice, we did not detect differences in co-stimulatory marker expression between *Stat3*^∆/∆^ and *Stat3^fl/fl^* CD103^+^ cDC1s in the TdLN. These data underscore the idea that tumor-associated STAT3-activating cytokines in the TME inhibit co-stimulatory molecule expression in CD103^+^ cDC1s, yet do not affect expression in the TdLN. Additional work is needed to understand the impact of discrete microenvironments (e.g., tumor, TdLNs) on STAT3 signaling in CD103^+^ cDC1s as well as CD103^+^ cDC1-mediated T cell activation and recruitment.

Immunotherapy has demonstrated limited efficacy in the treatment of breast cancer, and this has been linked with low amounts of CD8^+^ T cells in the TME [14,16]. Lack of CD8^+^ T cells in breast tumors may result in part from an abundance of immunosuppressive signals in the TME [14]. Consistently, our results as well as work from others indicates breast tumors express STAT3-activating cytokines [36]. Significantly, we show that STAT3 depletion in CD103^+^ cDC1s leads to increased CD8^+^ T cell infiltration in murine breast tumors, suggesting this approach overcomes the immunosuppressive breast TME. Thus, blockade of STAT3 in migratory cDC1s may be a novel means to overcome tumor immunosuppression and improve the efficacy of immunotherapies in human breast cancer. In addition, STAT3 inhibitors are currently in clinical trials for the treatment of many cancers [25]. Our data suggest these treatments may improve the ability of human non-lymphoid organ cDC1s to induce anti-tumor immunity and underscore the importance of evaluating tumor immune responses in STAT3 inhibitor clinical trials.

## 4. Materials and Methods

### 4.1. Mouse Strains

B6.Cg-Tg(Itgax-cre)1-1Reiz/J mice (CD11c Cre^+^) from the Jackson Laboratory (Bar Harbor, ME, USA) were bred with *Stat3^f/f^* mice to generate CD11c Cre^+^
*Stat3^f/f^* mice and CD11c Cre^−^
*Stat3^f/f^* littermate controls [48,49]. C57BL/6J, B6.SJL-*Ptprc^a^ Pep^b^*/BoyJ (CD45.1^+^), and B6.129S2-*Il10rb^tm1Agt^*/J (*Il10rb*^−/−^) mice were acquired from the Jackson Laboratory. All mice were maintained in a specific pathogen-free animal facility at UT MD Anderson Cancer Center and used in accordance with IACUC approved protocols.

### 4.2. In Vitro Generation of CD103^+^ cDC1s

Adapted from a protocol developed by Mayer et al. [26], murine bone marrow cells (BMs) were cultured in Roswell Park Memorial Institute (RPMI) 1640 medium (Thermo Fisher Scientific, Waltham, MA, USA) containing 10% heat-inactivated fetal bovine serum (FBS) (Atlanta Biologicals, Atlanta, GA, USA), 1 mM sodium pyruvate (Thermo Fisher Scientific), 50 μM β-mercaptoethanol (Thermo Fisher Scientific), and 1% penicillin-streptomycin (Thermo Fisher Scientific) (complete-RPMI), supplemented with 50 ng/mL human FLT3L (PeproTech, Rocky Hill, NJ, USA) and 2 ng/mL murine GM-CSF (PeproTech). BM cultures were initiated at a density of 1.5 × 10^6^ cells/mL and supplemented on d 5 with 5 mL of complete-RPMI per 10 mL of culture. On d 9, non-adherent cells were collected and transferred to fresh medium containing 50 ng/mL human FLT3L and 2 ng/mL murine GM-CSF at a density of 3 × 10^5^ cells/mL. On d 17, non-adherent cells were collected and CD103^+^ cDC1s (CD11c^+^ CD45R^−^ CD24^+^ CD172α^−^ CD103^+^) were purified by fluorescence activated cell sorting (FACS) on a FACSAria III or FACSAria Fusion (BD Biosciences, Palo Alto, CA, USA).

### 4.3. Administration of CD103^+^ cDC1 Vaccines In Vivo

FACS-purified CD103^+^ cDC1s were cultured at 4.5 × 10^6^ cells/mL in complete-RPMI supplemented with 20 μg/mL poly I:C (Millipore Sigma, Darmstadt, Germany), 400 μg/mL ovalbumin (OVA; a surrogate tumor antigen) (Millipore Sigma), and 20 ng/mL murine GM-CSF for 4 hours (h) at 37 °C. Cells were washed three times with phosphate-buffered saline (PBS) and resuspended in endotoxin-free PBS for injection into tumors. For studies evaluating tumor growth, mouse survival, or immune subsets post vaccination, 2 × 10^6^ purified CD103^+^ cDC1s were administered via i.t. injection on d 7 following tumor establishment. For studies examining CD103^+^ cDC1 vaccine migration and surface marker expression post vaccination, 5–7.5 × 10^6^ CD103^+^ cDC1s were injected i.t. on d 7 or d 14 following tumor establishment.

### 4.4. Murine Breast Cancer Cells and Establishment of Orthotopic Mammary Tumors

Murine breast cancer cells stably expressing OVA (PyMT-OVA) were kindly provided by Dr. David G. DeNardo and Dr. Melissa A. Meyer [36]. Briefly, the cell line was isolated from an end stage tumor in a PyMT transgenic C57BL/6 mouse and transduced to express OVA [36]. PyMT-OVA cells were cultured in Dulbecco’s modified eagle medium (DMEM) (Thermo Fisher Scientific) containing 10% fetal bovine serum and 1% penicillin-streptomycin. Cells were washed three times with PBS and resuspended in endotoxin-free PBS prior to injection into mice. Ten to twelve-week old female C57BL/6 mice received bilateral injections of 5 × 10^5^ PyMT-OVA cells in the mammary fat pad between the 4th and 5th mammary glands. Tumor length and width were measured every 2–3 d with electronic calipers. Mice were euthanized when tumors reached 15 mm in any direction or when ulceration >2 mm occurred.

### 4.5. Immune Cell Isolation from Tumors and Lymph Nodes

PyMT-OVA tumors were removed and cut into small pieces (~2 mm) with scissors. Tumor pieces were incubated in a digestion buffer containing 1 mg/mL collagenase type IV (Millipore Sigma, Darmstadt, Germany), 0.1 mg/mL hyaluronidase (Millipore Sigma), and 30 units/mL deoxyribonuclease (Millipore Sigma) in RPMI 1640 for 45 minutes (min) in a shaking incubator (Eppendorf New Brunswick Excella E25) at 37 °C and 100 RPM. Digested cell suspensions were passed through 100 μm mesh filters; cells were subsequently washed with PBS containing 2 mM ethylenediaminetetraacetic acid (EDTA) and 2% FBS (FACS buffer) in preparation for antibody staining. Inguinal TdLN were dissected; TdLN cells were passed through 100 μm mesh filters and washed with FACS buffer prior to antibody staining.

### 4.6. Immune Profiling by Antibody Staining and Flow Cytometry

Single-cell suspensions were incubated in FACS buffer containing rat anti-mouse CD16/32 antibody (Fc block, Tonbo Biosciences, San Diego, California, USA) for 15–30 min at 4 °C. Subsequently, samples were stained with fluorescently conjugated antibodies against murine cell surface markers for 20 min at 4 °C. For samples requiring analysis of intracellular proteins (T cell subsets), single-cell suspensions were incubated initially in complete-RPMI containing 0.5 μg/mL ionomycin (Millipore Sigma), 50 ng/mL phorbol 12-myristate 13-acetate (PMA, Millipore Sigma), and GolgiStop™ (BD Biosciences) for 4 h at 37 °C. Cell surface marker staining was followed by fixation and permeabilization according to manufacturer’s instruction (eBioscience Intracellular Fixation and Permeabilization Buffer Set, Thermo Fisher Scientific). After fixation and permeabilization, the cells were stained with fluorescently conjugated antibodies for intracellular proteins for 20 min at 4 °C. The following reagents were used: BV-421-conjugated CD11c (N418); APC-conjugated Ly6C (HK1.4), CD86 (GL-1), CD45R (RA3-6B2), or CD45.1 (A20) antibodies; redFluor™ 710-conjugated CD45.2 (104) antibody; APC-Cy7-conjugated CD4 (GK1.5) or CD11b (M1/70) antibodies; FITC-conjugated IFN-γ (XMG 1.2), Ly6G (1A8), CD172α (P84), or CD80 (16-10A1) antibodies; Percp-Cy5.5-conjugated CD3e (17A2), CD24 (M1/69), or F4/80 (BM8) antibodies; PE-conjugated FoxP3 (FJK-16S) or CD103 (2E7) antibodies; PE-Cy7-conjugated CD8α (53-6.7) or MHC II (M5/114.15.2) antibodies. All antibodies were purchased from Tonbo Biosciences, BD Biosciences, Thermo Fisher Scientific, or BioLegend (San Diego, CA, USA). OVA-specific CD8^+^ T cells were identified by staining with the SIINFEKL/H-2Kb-Pentamer (F093-4A, ProImmune, Sarasota, FL, USA). Dead cells were discriminated in all experiments using Ghost Dye™ violet 510 (Tonbo Biosciences). Stained single-cell suspensions were analyzed on a BD LSR Fortessa (BD Biosciences). Data analysis was performed using FlowJo v10 software (FlowJo, Ashland, OR, USA).

### 4.7. Cytokine Detection

CD103^+^ cDC1 supernatant cytokines were measured using the mouse ProcartaPlex panel 1A in accordance with manufacturer’s instruction (Invitrogen, Carlsbad, CA, USA) on a Luminex 200 machine (Luminex, Austin, TX, USA). Cytokines and chemokines analyzed: CXCL5, Eotaxin, G-CSF (granulocyte colony-stimulating factor), GM-CSF (granulocyte-macrophage colony-stimulating factor), CXCL1, IFN-α, IFN-γ, IL-1α, IL-1β, IL-10, IL-12p70, IL-13, IL-15/IL-15R, IL-17A, IL-18, IL-2, IL-22, IL-23, IL-27, IL-28, IL-3, IL-31, IL-4, IL-5, IL-6, IL-9, CXCL10, LIF, M-CSF (macrophage colony-stimulating factor), CCL2 (C-C motif ligand 2), CCL7, CCL3, CCL4, CCL5, and TNF-α (Tumor necrosis factor-α).

### 4.8. Statistics

All statistical analyses were performed using Prism 8 software (GraphPad Software, San Diego, CA, USA). Data are presented as the mean ± S.E.M. When comparing two groups, an unpaired, two-tailed *t* test was performed. When comparing more than two groups, one-way or two-way ANOVA with the indicated multiple comparisons test was performed. Differences were considered significant when *p* < 0.05.

## 5. Conclusions

Our data demonstrate that a CD103^+^ cDC1 based vaccine induces systemic anti-tumor immunity in murine breast cancer, a tumor type that is refractory to current immunotherapies. Furthermore, we reveal an inhibitory role for STAT3 and IL-10 in CD103^+^ cDC1 vaccine efficacy, as assessed by tumor growth and mouse survival. STAT3 negatively regulated CD103^+^ cDC1 maturation as well as vaccine promotion of CD8^+^ T cell and Th1 responses. Thus, our results suggest the potential for pharmacologic inhibition of STAT3 in CD103^+^ cDC1s as a novel treatment strategy for the immunotherapy of breast cancer.

## Figures and Tables

**Figure 1 cancers-12-00128-f001:**
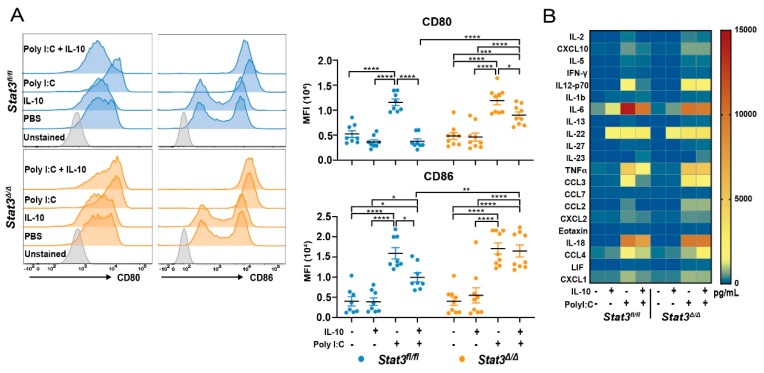
Evaluation of Signal Transducer and Activator of Transcription 3 (STAT3) function in CD103^+^ type 1 conventional dendritic cell (cDC1) maturation. In vitro-generated, fluorescence activated cell sorting (FACS)-purified *Stat3*-sufficient (Stat3^*fl/fl*^) or -deficient (*Stat3*^∆/∆^) CD103^+^ cDC1s were stimulated with or without 10 ng/mL murine interleukin (IL)-10, 20 μg/mL polyinosinic:polycytidylic acid (poly I:C), or 10 ng/mL IL-10 and 20 μg/mL poly I:C for 16 hours (h), as indicated. (**A**) Representative flow cytometry plots (left) and cumulative data (right) of cell surface expression of co-stimulatory markers on CD103^+^ cDC1s following the indicated treatment. *n* = 8 (*Stat3^fl/fl^*), *n* = 9 (*Stat3*^∆/∆^) for all conditions, combined from 4 independent experiments. (**B**) Mean cytokine and chemokine abundance in cell culture supernatants from (**A**), determined by multiplex analysis; cytokines and chemokines that were not detected within observable limits are not shown. *n* = 4 (phosphate buffered saline, PBS), *n* = 5 (IL-10), *n* = 6 (poly I:C), *n* = 6 (poly I:C + IL-10) for both genotypes, combined from 4 independent experiments. Data shown in panel (**A**) (right) represent mean ± S.E.M. (**A**) results analyzed by two-way ANOVA and Bonferroni’s multiple comparison test, comparing all treatments within each genotype, and comparing the two genotypes within a given treatment. Results were considered significant when *p* < 0.05. * *p* < 0.05; ** *p* < 0.01; *** *p* < 0.001; **** *p* < 0.0001.

**Figure 2 cancers-12-00128-f002:**
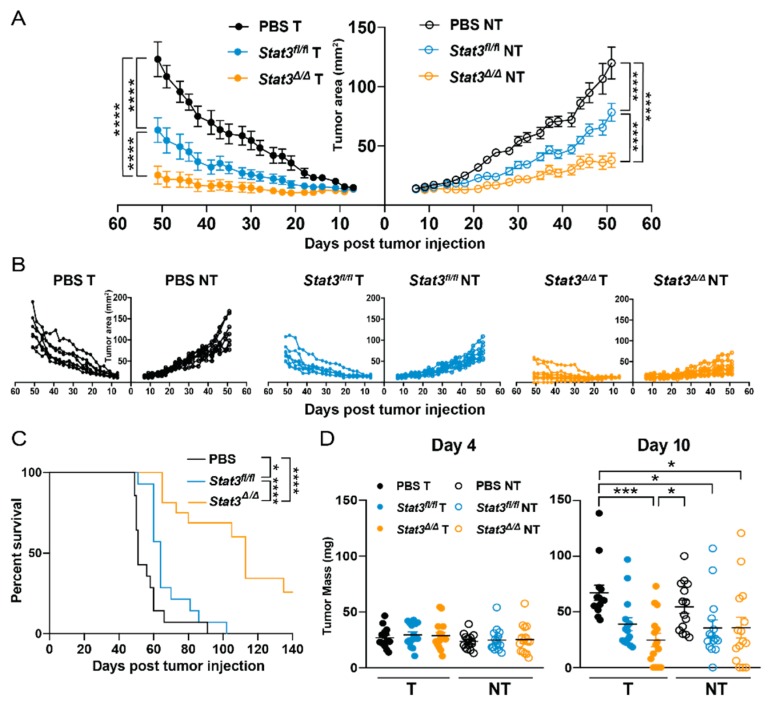
Efficacy of *Stat3*-deficient CD103^+^ cDC1 vaccine in murine breast cancer. Mice bearing bilateral Polyomavirus middle T-antigen (PyMT) tumors, engineered to overexpress ovalbumin (OVA, PyMT-OVA) received intratumoral (i.t.) injection of PBS, the *Stat3^fl/fl^* CD103^+^ cDC1 vaccine, or the *Stat3*^∆/∆^ CD103^+^ cDC1 vaccine in right side tumors only, 7 days (d) after tumor establishment. (**A**) Representative mean tumor size (mm^2^) of treated (T) and non-treated (NT) tumors, from one of two independent experiments. (**B**) Individual tumor sizes (mm^2^) from (**A**). (**A**,**B**) *n* = 7 (PBS), *n* = 7 (*Stat3^fl/fl^*), *n* = 8 (*Stat3*^∆/∆^). (**C**) Cumulative mouse survival from two independent experiments. *n* = 14 (PBS), *n* = 14 (*Stat3^fl/fl^*), *n* = 16 (*Stat3*^∆/∆^). (**D**) Tumor mass (mg) on the indicated day following i.t. PBS or CD103^+^ cDC1 vaccine injection; cumulative results from three independent experiments shown. Left panels, day 4; *n* = 15 (PBS T, NT; *Stat3^fl/fl^* T, NT; *Stat3*^∆/∆^ NT), *n* = 16 (*Stat3*^∆/∆^ T). Right panels, day 10; *n* = 14 (PBS T, NT), *n* = 15 (*Stat3^fl/fl^* T, NT; *Stat3*^∆/∆^ T, NT). (**A**,**D**) Data shown are the mean ± S.E.M. Results analyzed by two-way ANOVA (**A**) or one-way ANOVA and Tukey’s multiple comparisons test (**D**). (**C**) Data analyzed by log-rank (Mantel–Cox) test. Results were considered significant when *p* < 0.05. * *p* < 0.05; *** *p* < 0.001; **** *p* < 0.0001.

**Figure 3 cancers-12-00128-f003:**
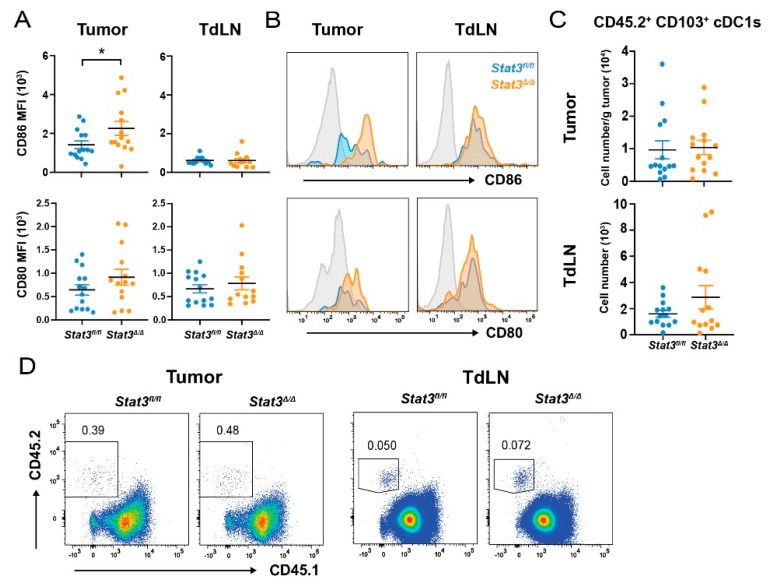
Maturation and migration of *Stat3*-deficient CD103^+^ cDC1s after intratumoral injection. Bilateral PyMT-OVA tumor-bearing CD45.1^+^ mice received i.t. injection of PBS, the CD45.2^+^
*Stat3^fl/fl^* CD103^+^ cDC1 vaccine, or the CD45.2^+^
*Stat3*^∆/∆^ CD103^+^ cDC1 vaccine in right side tumors only, 7 d after tumor establishment. The vaccine-derived CD45.2^+^ CD103^+^ cDC1s were identified in tumors and tumor-draining lymph nodes (TdLNs) by analysis of CD45.2^+^ cells 40 h after i.t. injection. (**A**) Cumulative and (**B**) representative cell surface expression of the indicated co-stimulatory molecules on vaccine-derived CD45.2^+^ CD103^+^ cDC1s. (**C**) Number of vaccine-derived CD45.2^+^ CD103^+^ cDC1s in tumors and TdLNs, as indicated. (**D**) Representative flow cytometry plots showing vaccine-derived CD103^+^ cDC1s as a percentage of live cells. (**A**,**C**) Data shown are the mean ± S.E.M. from two independent experiments. *n* = 14 (*Stat3^fl/fl^* tumor, TdLN; *Stat3*^∆/∆^ tumor), *n* = 13 (*Stat3*^∆/∆^ TdLN). Data analyzed by unpaired, two-tailed *t* test. Results were considered significant when *p* < 0.05. * *p* < 0.05.

**Figure 4 cancers-12-00128-f004:**
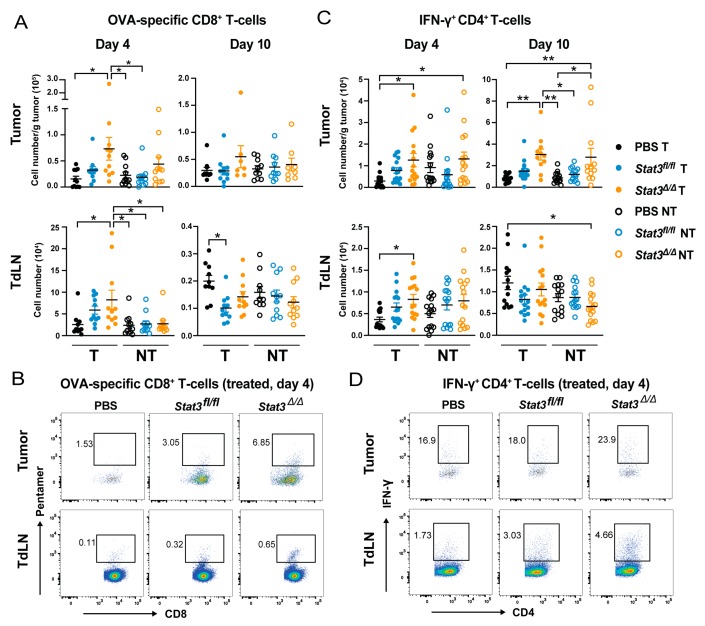
T cell responses after vaccination with *Stat3*-deficient CD103^+^ cDC1s. T cell subsets were analyzed in tumors and TdLNs 4 d and 10 d following i.t. vaccination of mice bearing bilateral PyMT-OVA tumors with *Stat3^fl/fl^* CD103^+^ cDC1s, *Stat3*^∆/∆^ CD103^+^ cDC1s or PBS. (**A**) Cumulative amounts and (**B**) representative flow cytometry plots of OVA-specific CD8^+^ T cells (identified by SIINFEKL/H2-Kb pentamer staining) (CD45^+^ CD3^+^ CD4^−^ CD8^+^ Pentamer^+^) within tumors and TdLNs on 4 d and 10 d following vaccination, as indicated. Data shown are from two independent experiments (**A**). Tumors, d 4, *n* = 11 for all groups; TdLNs, d 4, *n* = 11 (*Stat3^fl/fI^* T, NT; *Stat3*^∆/∆^ T; PBS NT), *n* = 10 (PBS T, *Stat3*^∆/∆^ NT). Tumors, d 10, *n* = 10 (PBS T, NT; *Stat3^fl/fI^* NT), *n* = 11 (*Stat3^fl/fl^* T), *n* = 7 (*Stat3*^∆/∆^ T), *n* = 8 (*Stat3*^∆/∆^ NT). TdLNs, d 10, *n*= 10 (PBS T, NT), *n* = 11 (*Stat3^fl/fl^* T, NT; *Stat3*^∆/∆^ T, NT). (**C**) Cumulative amounts and (**D**) representative flow cytometry plots of IFN-γ^+^ CD4^+^ T cells (CD45^+^ CD3^+^ CD8^−^ CD4^+^ IFN-γ^+^) within tumors and TdLNs on 4 d and 10 d following vaccination, as indicated. Data shown are from three independent experiments (**C**). Tumors, d 4, *n* = 15 (PBS T, NT; *Stat3^fl/fl^* NT; *Stat3*^∆/∆^ T, NT), *n* = 16 (*Stat3^fl/fl^* T). TdLNs, d 4, *n* = 15 for all groups. Tumors, d 10, *n* = 14 (PBS T, NT), *n* = 15 (*Stat3^fl/fl^* T), *n* = 11 (*Stat3*^∆/∆^ T), *n* = 13 (*Stat3^fl/fl^* NT), *n* = 12 (*Stat3*^∆/∆^ NT). TdLNs, d 10, *n* = 13 (PBS T), *n* = 14 (PBS NT), *n* = 15 (*Stat3^fl/fl^* T, NT; *Stat3*^∆/∆^ T, NT) (**A**,**C**) Data shown are the mean ± S.E.M. Data analyzed by one-way ANOVA and Tukey’s multiple comparisons test. Results were considered significant when *p* < 0.05. * *p* < 0.05; ** *p* < 0.01.

**Figure 5 cancers-12-00128-f005:**
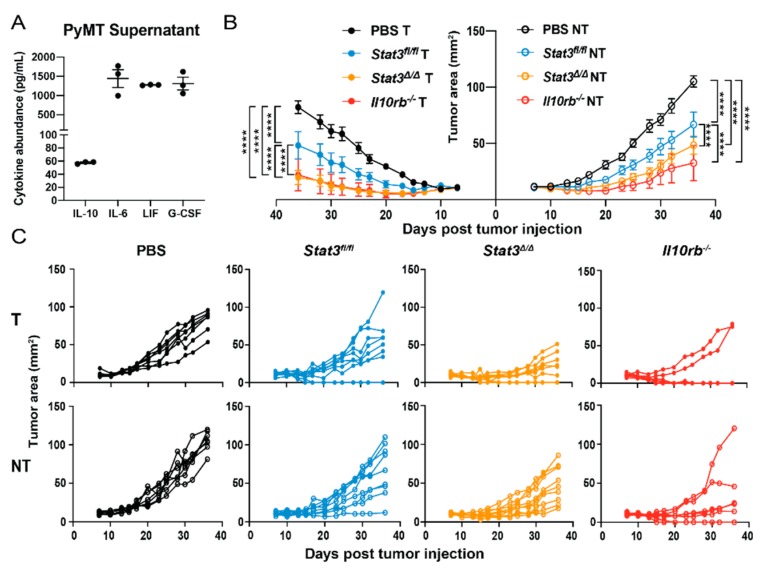
Efficacy of IL-10 receptor β subunit-deficient (*Il10rb*^−/−^) CD103^+^ cDC1 vaccine in murine breast cancer. (**A**) Cytokine amounts in PyMT-OVA culture supernatant determined by multiplex analyses. Data are from three independent experiments, *n* = 3. (**B**,**C**) Bilateral PyMT-OVA tumor-bearing mice received i.t. injection of PBS, the *Stat3^fl/fl^* CD103^+^ cDC1 vaccine, the *Stat3*^∆/∆^ CD103^+^ cDC1 vaccine, or the *Il10rb*^−/−^ CD103^+^ cDC1 vaccine in right side tumors only, 7 d after tumor establishment. (**B**) Representative mean tumor size (mm^2^) from one of two independent experiments. (**C**) Individual tumor sizes from (**B**), *n* = 7 (PBS, *Il10rb*^−/−^), *n* = 9 (*Stat3^fl/fl^*, *Stat3*^∆/∆^). (**A**,**B**) Data shown are the mean ± S.E.M. (**B**) Data analyzed by two-way ANOVA and Tukey’s multiple comparisons test. Results were considered significant when *p* < 0.05. **** *p* < 0.0001.

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
