# Peer review of "STAT3 Inhibits CD103+ cDC1 Vaccine Efficacy in Murine Breast Cancer"

_cancers, 2020, doi:10.3390/cancers12010128_

Round 1
Reviewer 1 Report
In this study the authors described the negative regulation through STAT3 of poly-IC induced CD103+ DC activation, leading to down-modulated anti-tumor activity.
The quality of the work is of high level and the results are convincing and nicely presented. One important issue that the author may have the opportunity to address concern the real site of STAT3 mediated DC inhibition (in vivo upon IT injection, or in vitro following Poly-IC activation).
Major comments:
Link to the above major comment: Fig.5 reports high levels of IL6, G-CSF and LIF in the tumor supernatant, while very little IL10 is detected. This in sharp contrast with the therapeutic activity of IL10Rb-/- DC. This may argue for an impact of STAT3 and IL10 during the in vitro generation rather than in vivo. The SN from the DC should be measured for the same cytokines In this same line, it would be good to design an experiment where endogenous DC will be deficient for STAT3. Can the tumor model be implanted in STAT3-/- mice and IL10Rb-/-, ideally in the myeloid compartment (or even a complete KO) and treated with Poly-IC IT. For cDC1 generation it is needed to provide either reference to previous work or to show the gating strategy for the Facs-sorting of the in vitro generated cDC1. As the paper is mentioned as submitted, the manuscript should be provided or data shown as supplementary: Zhou, Y., N. Slone, T. T. Chrisikos, O. Kyrysyuk, R. Babcock, Y. B. Medik, H. S. Li, E.S. Kleinerman, and 561 S. S. Watowich. "Efficacy of a Murine CD103+ Conventional Dendritic Cell Vaccine against Primary and 562 Metastatic Cancer." Submitted (2019).
Minor comments
Issue with Terminology: CD103+ enough to call them cDC1?, need to document XCR1, absence of SIRPa intratumor injection of DC vaccine: terminology of DC vaccine not appropriate as DC are not loaded with ag source. Ex-vivo DC is not appropriate, they are in vitro generated. 2- Fig.3: a bit surprising that DC in the draining LN are less activated than the tumor DC
Reviewer 2 Report
This manuscript dose an excellent job in demonstrating inhibitory role of STAT3 in regulating CD103+ cDC1 vaccine efficacy. Experiments were well designed and results are clearly presented. To demonstrate the efficacy of CD103+ cDC1 vaccine and role of STAT3, authors exploited the well described murine PyMT-OVA BC model. It might be interesting and relevant to investigate the efficacy of this STAT3 deficient cDC1 vaccine in other murine BC models, e.g. a more clinical relevant inflammatory BC model.
